# A Method for the Analysis of Glyphosate, Aminomethylphosphonic Acid, and Glufosinate in Human Urine Using Liquid Chromatography-Tandem Mass Spectrometry

**DOI:** 10.3390/ijerph19094966

**Published:** 2022-04-19

**Authors:** Zhong-Min Li, Kurunthachalam Kannan

**Affiliations:** 1Department of Pediatrics, New York University Grossman School of Medicine, New York, NY 10016, USA; zhongmin.li@nyulangone.org; 2Department of Environmental Medicine, New York University Grossman School of Medicine, New York, NY 10016, USA; 3King Fahd Medical Research Center, Biochemistry Department, Faculty of Science and Experimental Biochemistry Unit, King Abdulaziz University, Jeddah 80200, Saudi Arabia

**Keywords:** glyphosate, aminomethylphosphonic acid, glufosinate, urine, LC–MS/MS

## Abstract

The extensive use of herbicides, such as glyphosate and glufosinate, in crop production during recent decades has raised concerns about human exposure. Nevertheless, analysis of trace levels of these herbicides in human biospecimens has been challenging. Here, we describe a method for the determination of urinary glyphosate, its degradation product aminomethylphosphonic acid (AMPA), and glufosinate using liquid chromatography-tandem mass spectrometry (LC–MS/MS). The method was optimized using isotopically labelled internal standards (^13^C_2_, ^15^N-glyphosate, ^13^C, ^15^N, D_2_-AMPA, and D_3_-glufosinate) and solid-phase extraction (SPE) with cation-exchange and anion-exchange cartridges. The method provides excellent chromatographic retention, resolution and peak shape of target analytes without the need for strong acidic mobile phases and derivatization steps. The instrument linearity was in the range of 0.1–100 ng/mL, with *R* > 0.99 in the matrix for all analytes. The method detection limits (MDLs) and the method quantification limits (MQLs) were in the ranges of 0.12 (AMPA and glufosinate)–0.14 (glyphosate) ng/mL and 0.40 (AMPA)–0.48 (glyphosate) ng/mL, respectively. The recoveries of analytes spiked into urine matrix ranged from 79.1% to 119%, with coefficients of variation (CVs) of 4–10%. Repeated analysis of samples for over 2 weeks showed intra-day and inter-day analytical variations of 3.13–10.8% and 5.93–12.9%, respectively. The matrix effects for glyphosate, AMPA, and glufosinate spiked into urine matrix averaged −14.4%, 13.2%, and 22.2%, respectively. The method was further validated through the analysis of external quality assurance proficiency test (PT) urine samples. The method offers optimal sensitivity, accuracy, and precision for the urine-based assessment of human exposure to glyphosate, AMPA, and glufosinate.

## 1. Introduction

Glyphosate (*N*-(phosphonomethyl)glycine) and glufosinate (2-amino-4-(hydroxy(methyl)phosphoryl)butanoic acid) are non-selective, broad-spectrum herbicides used in both agricultural and non-agricultural sectors. Their use in agriculture has greatly increased since the development of crop strains genetically modified to tolerate them. The current annual use of glyphosate, the most widely used herbicide, is estimated at 600,000–750,000 tons of active ingredients and is expected to increase to 740,000–920,000 tons by 2025 [1]. The United States accounts for 19% of the global glyphosate usage and >100,000 tons of glyphosate have been applied annually in agriculture since 2010 [2]. Glufosinate is mainly used to control glyphosate-resistant weeds. The United States is the world’s largest market for glufosinate with >4000 tons of active ingredients used annually since 2016 [3]. The large-scale use of glyphosate and glufosinate has resulted in their ubiquitous presence in food and environmental matrices [4,5,6].

Concerns over human exposure to glyphosate and its analogues are mounting due to potential health risks [7,8]. Studies have reported human exposure to glyphosate, its degradation product aminomethylphosphonic acid (AMPA), and glufosinate through food and water [9,10]. After ingestion, these compounds are primarily excreted unchanged in feces and urine [11]. Human biomonitoring studies have reported the occurrence of glyphosate and AMPA in urine collected from different countries, with average glyphosate concentrations ranging between 0.26 and 73.5 ng/mL in occupationally exposed individuals and between 0.16 and 7.6 ng/mL in the general population [9]. The acute oral toxicity of glyphosate in rats was low, with LD50 values greater than 5000 mg/kg-bw. The health risks from exposure to glyphosate include oxidative stress [12], anti-estrogenicity [13], anti-androgenicity [13], reproductive toxicity [14], and carcinogenicity [15]. For glufosinate, the acute oral and dermal toxicity was low, with LD50 values greater than 1500 mg/kg-bw in rats and mice [16]. The European Food Safety Authority concluded that there was no evidence for genotoxicity, carcinogenicity, or neurotoxicity of glufosinate. However, a reversible reduction in glutamine concentration in mammalian tissues was observed following exposure to high levels of glufosinate [17]. Additionally, glufosinate was shown to induce pre- and post-implantation loss, vaginal bleeding, abortion and dead fetus in rats [18].

The International Agency for Research on Cancer (IARC) classified glyphosate as a probable human carcinogen (Group 2A) [19]. Nevertheless, controversies surround its carcinogenic potential [10,20,21]. There exists a need for additional research regarding human exposure to glyphosate, AMPA, and glufosinate, especially in the general population.

Accurate determination of urinary glyphosate, AMPA, and glufosinate is hampered by these chemicals’ high polarity, hydrophilicity, and low molecular weights. The high polarity affects chromatographic separation, and the low molecular weight, in a range where interferences are frequent, affects test specificity (creating a high risk for false positives). Liquid chromatography (LC) coupled with UV detection (LC–UV) [22], LC with fluorescence detection [23], and gas chromatography coupled with mass spectrometry (GC–MS) [8] have been used to analyze glyphosate in environmental samples. However, these methods require derivatization of the analytes, a tedious and time-consuming step involving toxic reagents. More recently, with the advent of the more sensitive and rugged liquid chromatography-tandem mass spectrometry (LC–MS/MS) method, it has become possible to determine glyphosate without a derivatization step. Different strategies have been used to improve chromatographic retention and peak shape, including cation-exchange [24,25], anion-exchange [26], and ion-pairing reversed-phase chromatography [27]. However, poor peak shape, lack of adequate sensitivity, and the use of strongly acidic mobile phases to enhance ionization (e.g., ≥1% formic or acetic acids) hamper the analysis of trace levels of these chemicals in human specimens such as urine, especially for application in large-scale human biomonitoring studies [26,27,28,29,30]. Furthermore, despite its significance as a widely used herbicide, glufosinate has rarely been measured in human specimens [31,32].

Our aim was to develop a method for sensitive and selective determination of urinary glyphosate, AMPA, and glufosinate using isotope dilution LC–MS/MS suitable for application in large-scale human biomonitoring studies. We optimized the method to improve chromatographic retention and peak shape, eliminate matrix effects, and improve sensitivity while using milder mobile phases (less corrosive conditions) and avoiding derivatization. We then validated its sensitivity, accuracy, precision, and matrix effects by using fortified human urine samples and analyzing external quality assurance proficiency test (PT) urine samples.

## 2. Materials and Methods

### 2.1. Reagents and Materials

The molecular structures of the target analytes are shown in Figure 1. Glyphosate (10 µg/mL in water), ^13^C_2_,^15^N-glyphosate (100 µg/mL in water), AMPA (100 µg/mL in water), and ^13^C,^15^N,D_2_-AMPA (100 µg/mL in water) with purities of 95–98% were purchased from Cambridge Isotope Laboratories (Andover, MA, USA). Glufosinate and D_3_-glufosinate (purity ≥ 95%) were from Toronto Research Chemicals (Toronto, ON, Canada). Primary stock solutions of glufosinate and D_3_-glufosinate (1 mg/mL) were prepared in water. Working standard solutions were diluted from stock solutions using water:acetonitrile (ACN) (95:5, *v*/*v*) containing 0.1% formic acid. Formic acid (88%) and ammonium hydroxide (NH_4_OH; 28–30%) of analytical grade were obtained from Sigma-Aldrich (St. Louis, MO, USA). LC passivation solution containing 10 M medronic acid was from Restek Corp (Bellefonte, PA, USA). Water, methanol (MeOH), and ACN were purchased from Fisher Scientific (Waltham, MA, USA). Oasis^®^ MAX cartridges (60 mg/3 mL) and Oasis^®^ MCX cartridges (60 mg/3 mL) were obtained from Waters Corp. (Milford, MA, USA).

A small number of archived human urine samples previously collected for other studies were analyzed [33]. Institutional Review Board approvals were obtained from New York State Department of Health for the analysis of de-identified urine samples (under exempt category) to demonstrate application of the method developed in this study.

### 2.2. Sample Preparation

A 250 µL aliquot of each urine sample was transferred into a 15 mL polypropylene (PP) tube. Urine samples were fortified with the target compounds and internal standards at 0.5, 1, and 5 ng/mL concentrations (in water: ACN [95:5 *v*/*v*] containing 0.1% formic acid) for method optimization and validation. The sample was vortexed vigorously and kept at room temperature for 30 min. The mixture was loaded onto an Oasis MCX cartridge that had been preconditioned with 2 mL MeOH and 2 mL water. The eluate was collected immediately, as the target analytes were not absorbed by the cation-exchange cartridges (this step was for purification and removal of cationic interferences). The cartridge was then washed with 2 mL water, and the eluate was collected and combined. Thereafter, 2.5 mL of 3% NH_4_OH (*v*/*v*) aqueous solution was added and vortexed vigorously. The mixture (~5 mL in total) was then loaded onto an Oasis MAX cartridge preconditioned with 2 mL MeOH, 2 mL water, and 1 mL of 3% NH_4_OH. The cartridge was washed with 2 mL of 3% NH_4_OH and 2 mL MeOH, and moisture was removed using a vacuum pump for 3 min. The analytes were then eluted into a 15 mL PP tube with 3 mL of 3% formic acid in MeOH (*v*/*v*), and the eluate was evaporated to dryness under N_2_ at 40 °C. The residue was reconstituted in 250 µL of water: ACN (95:5, *v*/*v*) containing 0.1% formic acid, vortexed vigorously, and transferred into a glass vial. Finally, 20 µL of the sample was injected into the LC–MS/MS instrument.

### 2.3. LC–MS/MS

Identification and detection of the target analytes were performed using an AB Sciex 5500 Q-trap mass spectrometer (Framingham, MA, USA) coupled with a Shimadzu LC-30 AD ultra-high-performance liquid chromatograph (Shimadzu Corp., Kyoto, Japan). Analytes were separated on a Gemini^®^ C6-Phenyl column (150 × 4.6 mm, 5 µm; Phenomenex, Torrance, CA, USA) connected to a Betasil C18 guard column (20 × 2.1 mm, 5 µm; Thermo Fisher Scientific, Waltham, MA, USA). The mobile phases were water (A) and ACN (B) each containing 0.1% formic acid (*v*/*v*). The following mobile-phase gradient program was used: hold at 5% B for 2 min, linear ramp to 95% B over 8 min, hold at 95% B for 1 min, then return to initial conditions in over 1 min, and equilibrate at initial conditions for additional 2 min prior to the next injection. The column temperature was maintained at 40 °C; the autosampler temperature was 15 °C; and the mobile phase flow rate was 0.8 mL/min.

The target analytes were determined using negative-ion electrospray ionization (ESI) in the multiple reaction monitoring (MRM) mode. The MRM parameters, including declustering potential (DP), collision energy (CE), and collision cell exit potential (CXP), are shown in Appendix A. The IonSpray voltage was −5.5 kV; the ionization source temperature was 500 °C; and the curtain gas flow rate was 20 psi. Data were acquired and processed using the Analyst software, version 1.7.2 (AB Sciex, Framingham, MA, USA). Typical MS/MS chromatograms of the target compounds in standard solution are shown in Figure 2.

### 2.4. Method Validation

The method was validated by following a protocol of the New York State Department of Health (Wadsworth Center, Laboratory of Organic Analytical Chemistry; available at: https://www.wadsworth.org/sites/default/files/WebDoc/NYS%20DOH%20MML-301-06SOP.pdf (accessed on 10 March 2022)). Calibration curves were constructed for standards prepared both in neat solution and in fortified urine matrix. Calibration standards ranged in concentrations from 0.05 to 100 ng/mL, with 10 ng/mL of labelled internal standards, diluted from stock solutions with HPLC-grade water: ACN (95:5, *v*/*v*) containing 0.1% formic acid. Matrix-matched calibration curves were prepared by spiking various concentrations of the target analytes (0.1, 0.2, 0.5, 1, 2, 5, 10, 20, 50, and 100 ng/mL) into pooled urine.

Matrix effect was calculated as the percentage of signal enhancement or suppression, as shown in Equation (1):Matrix effect (%) = (A/B − 1) × 100(1)
where A and B are the slopes of analytes from the matrix-matched calibration curve and calibration curve prepared in neat solution, respectively.

The instrument detection limit (IDL) and instrument quantification limit (IQL) were defined as the concentrations of analytes in solvent that produced a peak with a signal-to-noise ratio (*S*/*N*) of 3 and 10, respectively. To estimate the method detection limit (MDL) and method quantification limit (MQL), six pooled urine samples were fortified with each target analyte individually at 0.5 ng/mL, a concentration that yielded peaks with *S*/*N* values of 11.3, 7.3, and 12.8 for glyphosate, AMPA, and glufosinate, respectively. MDL and MQL were calculated as 3 and 10 times the standard deviation (SD) measured in matrix, spiked at 0.5 ng/mL, respectively.

The accuracy of the method was determined as the recoveries of analytes spiked at three different concentrations (0.5, 1 and 5 ng/mL) in pooled urine. Procedural blank samples (water in place of urine) were included to monitor for background levels contamination. The precision of the method was assessed by intra-day and inter-day variations, which were calculated as the percentage of the coefficient of variation (%CV) of the measured concentrations in six pooled urine samples spiked at 0.5, 1, and 5 ng/mL, respectively. The inter-day CV was measured by repeated injection of fortified samples over a period of 2 weeks.

## 3. Results and Discussion

### 3.1. Chromatography and Mass Spectrometry

Reported LC–MS/MS methods for the determination of glyphosate and AMPA in urine are summarized in Table 1. Due to the highly polar and hydrophilic nature of the target analytes, chromatographic retention and separation using conventional reversed-phase columns (e.g., C18 column) is arduous, resulting in their co-elution with other matrix components. Retention of such analytes can be improved by reversed-phase ion-pair chromatography [27], which is based on the addition of ion pair reagents in the mobile phase to promote the formation of ion pairs. The increase in the hydrophobic character of the electrically neutral ion pair results in a greater affinity for the reverse stationary phase. Because of their strong hydrophobic interactions, the ion pair reagents cannot be completely flushed out of the LC column even through extensive washing, and thus require the use of a dedicated column for a particular application. Hydrophilic interaction liquid chromatography (HILIC) columns enable the retention and separation of hydrophilic compounds, but they often lead to poor peak shape due to interactions with metals in the stationary phase or the chromatographic hardware [34]. Alternatively, considering the low pKa values of the analytes (0.8 for the first phosphonate of glyphosate, 0.9 for the first phosphonate of AMPA, and 0.8 for the phosphonate of glufosinate [26,30]), an anion-exchange column was expected to offer efficient retention. However, a high concentration of an acid (e.g., 1% formic acid) was needed in the mobile phases to maintain an optimal peak shape [26]. Other studies have employed a cationic (−H^+^) guard column. Although this resulted in separation, glyphosate was eluted within a short retention time, while the peak shape of AMPA was poor [35,36]. In this study, we compared the performance of different chromatographic columns, including reversed-phase (C18-, C8-, C6-Phenyl), HILIC, and anion-exchange (polymer-based NH_2_, hydroxide-selective anion-exchange) columns (data not shown), and found that the C6-Phenyl column exhibited the best chromatographic performance. All analytes were well separated, and the peak shape of AMPA and glufosinate was symmetrical (Figure 2 and Figure 3). However, peak tailing was observed for glyphosate (Figure 3), probably due to the chelation of glyphosate with metal ions in the LC system [37]. Hsiao et al. recommended the addition of 5 µM medronic acid in the mobile phase (passivation solution) to eliminate chelation by metal ions and improve peak shape for metal-sensitive compounds [37]. Nevertheless, we observed a reduced intensity (by ~2-fold) for all analytes when medronic acid was added in mobile phases, indicative of ionization suppression. As an alternative, we passivated the LC system by injecting 10 mM medronic acid before analyzing real samples, i.e., we injected 20 µL of 10 mM medronic acid at the beginning of the analytical run (with the mobile phases directed to waste instead of the mass spectrometer). After this passivation, no ionization suppression was found and all analytes including glyphosate exhibited sharp and symmetrical peaks (Figure 3) for at least 300 subsequent injections.

### 3.2. Optimization of Sample Cleanup

Because of their low pKa values, we expected anion-exchange cartridges, which are positively charged and can bind negatively charged target analytes, to be effective for this application [26,38]. We first optimized a mixed-mode anion-exchange cartridge (Oasis^®^ MAX cartridge), which contains sorbents having both hydrophobic and anion-exchange functionalities. Indeed, MAX cartridges provided excellent recoveries for all target analytes after optimization of elution solvents. However, matrix components were not completely removed, as we observed strong ionization suppression of glyphosate and glufosinate, which resulted in poor sensitivity. For example, the *S*/*N* values of glyphosate and glufosinate in pooled urine spiked at 0.5 ng/mL were <3 and 3.8, respectively (Appendix A). Therefore, we introduced an additional purification step to reduce matrix effects. We compared several cartridges for cleanup, including reversed-phase cartridges (hydrophilic lipophilic balanced (HLB) solid-phase extraction (SPE) cartridges, C18, and graphitized non-porous carbon) and mixed-mode strong cation-exchange cartridges (Oasis^®^ MCX) (data not shown). We found that a pre-cleanup step in which samples were passed through MCX cartridges (as described above) significantly reduced matrix effects, and thus increased the method sensitivity. The responses of all analytes, especially that of glyphosate, increased considerably after MCX pre-cleanup (Appendix A). The peak area of glyphosate was >10-fold higher in urine sample passed through MCX and MAX cartridges than in those that passed only through MAX (Appendix A). These results highlighted the efficacy of MCX SPE as a pre-cleanup step for the improvement of method sensitivity. We believe that this is due to the efficient removal of cationic components from the matrix. An earlier study reported the successful use of MCX cartridges for pre-cleanup in the analysis of glyphosate and AMPA in foodstuffs [38]. LC–MS/MS chromatograms obtained following a combination of cation-exchange and anion-exchange SPE cartridges in the preparation of urine samples showed well-resolved peaks in samples fortified at 0.5 ng/mL (Figure 4).

### 3.3. Method Validation

We assessed the linearity of the instrument by injecting analytical standards prepared both in solvent (0.05–100 ng/mL) and urine matrix (0.1–100 ng/mL). An excellent linearity was found for all analytes with *R* values >0.99 (Table 2). We assessed the accuracy of the method using the recoveries of analytes fortified at three different concentrations (0.5, 1 and 5 ng/mL) in a pooled urine matrix and analyzed in six replications. The recoveries of glyphosate, AMPA, and glufosinate were 79.1–84.4% (mean: 81.6%), 100–109% (mean: 103%), and 106–119% (mean: 112%), respectively, with CV values of 8.4–9.6% (mean: 9.3%), 4–8% (mean: 6%), and 5–10% (mean: 8%), respectively (Table 2). We also assessed the intra-day and inter-day precision of the method by analyzing fortified samples (0.5, 1 and 5 ng/mL) repeatedly for six times over a period of two weeks. The intra-day CVs were 3.13–8.83% (mean: 6.38%), 3.19–10.8% (mean: 7.70%), and 3.46–10.1% (mean: 6.06%) for glyphosate, AMPA, and glufosinate, respectively, and the inter-day CVs were 7.22–9.09% (mean: 8.52%), 5.93–7.85% (mean: 7.10%), and 6.61–12.9% (mean: 10.0%), respectively (Table 2).

We determined the sensitivity of the method as IDLs/IQLs as well as MDLs/MQLs through the injection of standards and fortified urine samples. The respective IDLs and IQLs were 0.01 and 0.05 ng/mL for all target analytes. The MDLs/MQLs were 0.14/0.48, 0.12/0.40, and 0.12/0.41 ng/mL for glyphosate, AMPA, and glufosinate, respectively. The sensitivity of our method is comparable to those found in several previous studies [11,39,40,41,42], and slightly higher than those of others [27,43] (Table 1). We expect that further improvements in MDLs/MQLs could be accomplished through inclusion of additional sample volumes available for extraction.

**Table 1 ijerph-19-04966-t001:** Reported analytical methods for the measurement of glyphosate, AMPA, and glufosinate in urine.

Sample Type	Analytes	Internal Standards	Sample Cleanup	LC Condition	MS/MS	LODs/LOQs (ng/mL)	Ref(s).
*Cation-exchange column*							
Human urine	Glyphosate, AMPA	D_2_,^13^C_3_-Glyphosate; ^13^C,^15^N-AMPA	SPE cleanup using Oasis HLB cartridges (3 cc, 60 mg)	Bio-Rad Micro-Guard Cation-H^+^ column (30 × 4.6 mm, 9 µm); A: water B: 0.2% formic acid in ACN	Glyphosate: *168*/*63*, 168/150; AMPA: 110/79, *110*/*63*;	IDL: 0.02–0.04 IQL: 0.05–0.1 ^a^	[25]
Human urine	Glyphosate, AMPA	^13^C_3_,^15^N-Glyphosate; D_2_,^13^C^15^N-AMPA	Diluted with 0.1% formic acid, shaken and centrifuged	Bio-Rad Micro-guard Cation-H^+^ column (30 × 4.6 mm, 9 µm); A: 0.1% formic acid in water B: ACN	Glyphosate: *168*/*63*, 168/79 AMPA: *110*/*63*, 110/79	MDL: 0.023–0.041 MQL: 0.1	[36,44]
Human urine	Glyphosate, AMPA	^13^C_2_,^15^N-Glyphosate; D_2_,^13^C,^15^N-AMPA	Refer to Jensen et al. [36]		Glyphosate: *168*/*63*, 168/126 AMPA: *110*/*63*, 110/79	MDL: 0.05–0.09 MQL: 0.20	[45]
*Anion-exchange column*							
Pet urine (dogs and cats)	Glyphosate, AMPA	^13^C_2_,^15^N-Glyphosate; D_2_,^13^C,^15^N-AMPA	(1) Sample basified with 1% NH_4_OH; (2) Cleanup using Oasis MAX SPE cartridge (3 cc, 60 mg)	Dionex IonPac AS21 IC column (250 × 2.0 mm, 7 µm); Isocratic elution: 1% formic acid in ACN/water (5/95)	Glyphosate: *168*/*63*, 168/79; AMPA: *110*/*63*, 110/79;	MDL: 0.15 ^a^ MQL: 0.5	[26]
Human urine	Glyphosate	^13^C_2_,^15^N-Glyphosate	Sample diluted with 1% formic acid, then filtered	Dionex IonPac AS 21 (250 × 2.0 mm, 7 µm); Isocratic elution: 1% formic acid in ACN/water (5:95)		MDL: 0.1 ^a^ MQL: 0.33	[46]
*Hybrid-phase column*							
Human urine	Glyphosate	^13^C_2_-*N*-Glyphosate	−	Obelisc-N mixed-mode column (100 × 2.1 mm, 5 µm); Isocratic elution: 1% formic acid in water	*168*/*63*, 168/81	MDL: 0.1 MQL: 0.5	[47]
*Reversed-phase column*							
Human urine	Glyphosate	^13^C_2_,^15^N-Glyphosate	(1) Sample diluted with water; (2) SPE: Strata SAX (1 cc, 100 mg)	Zorbax SB-C3 column (150 × 4.6 mm, 5 µm), or Zorbax XDB-C8 column (150 × 4.6 mm, 5 µm) A: 1% acetic acid in water B: ACN	168/63	MQL: 0.5	[11,40,41,42]
Human urine	Glyphosate	^13^C_2_,^15^N-Glyphosate	(1) Sample diluted with H_2_O; (2) SPE: ISOLUTE-96 SCX plate (25 mg), then ISOLUTE-96 NH_2_ plate (100 mg)	Scherzo SM-C18 MF column (100 × 2 mm, 3 µm) A: MeOH/water (5:95) containing 0.1% formic acid and 5 µM medronic acid B: MeOH and 20 mM ammonium formate (20:80) with 5 µM medronic acid	*170*/*88*, 170/60, 170/42 ^b^	MDL: 0.1 MQL: 0.3	[39]
Human urine	Glyphosate	−	−	SUPELCO Discovery C18 column (50 × 2.1 mm, 5 µm)	−	MDL: 1 MQL: 2	[43]
*Reversed-phase column (Ion-pairing chromatography)*							
Human urine	Glyphosate, Glufosinate	^13^C_2_,^15^N-Glyphosate; D_3_-Glufosinate	(1) Dilute with water; (2) Back wash with dichloromethane	Agilent ZORBAX SB-Aq column (100 × 2.1 mm, 1.8 µm) A: 15 mM HFBA; B: ACN	Glyphosate: *170*/*88*, 170/60; Glufosinate: *182*/*136*, 182/119	MDL: 0.1	[31]
Human urine	Glyphosate, AMPA	^13^C_3_,^15^N-Glyphosate; ^13^C,^15^N-AMPA	Sample diluted with HFBA	Gemini C6-Phenyl column (150 × 4.6 mm, 5 µm) A: 15 mM HFBA in water B: ACN	Glyphosate: *170*/*88*, 170/60; AMPA: 112/30 ^b^	MDL: 2.5 MQL: 5	[27]
*Reversed-phase column (derivatization)*							
Human urine	Glyphosate. AMPA, Glufosinate	^13^C_3_,^15^N-Glyphosate; D_2,_^13^C,^15^N-AMPA; D_3_-Glufosinate	(1) EDTA pre-treatment; (2) SPE: Strata-X; (3) Derivatization; (4) SPE: C18	Kinetex C18 column A: 5 mM AmAc (pH 9):MeOH:ACN (90:5:5) B: MeOH: ACN (50:50)	ESI positive, SIM mode Glyphosate-Fmoc: 392.08937 AMPA-Fmoc: 334.083890 Glufosinate-Fmoc: 404.12575	MDL: 0.1–0.3	[32]

Abbreviations: ACN, acetonitrile; AMPA, aminomethylphosphonic acid; HFBA, heptafluorobutyric acid; HLB, hydrophilic-lipophilic balanced; MDL, method detection limit; MQL, method quantification limit; SPE, solid-phase extraction; SIM, selective ionization mode; F-moc: 9-fluorenylmethoxycarbonyl chloride. ^a^ The authors did not specify whether the values are instrument detection limits/instrument quantification limit (IDL/IQLs) or MDL/MQLs. ^b^ Analytes were measured under ESI positive-ionization mode. Italicized transitions indicate quantitative ions monitored.

**Table 2 ijerph-19-04966-t002:** Optimized analytical parameters for the analysis of glyphosate, AMPA, and glufosinate in human urine. AMPA, aminomethylphosphonic acid; IDL, instrument detection limit; IQL, instrument quantification limit; MDL, method detection limit; MQL, method quantification limit.

	Glyphosate	AMPA	Glufosinate
*R* in solvent ^a^	0.9995	0.9999	0.9999
*R* in matrix ^b^	0.9982	0.9993	0.9998
IDL (ng/mL)	0.01	0.01	0.01
IQL (ng/mL)	0.05	0.05	0.05
MDL (ng/mL)	0.14	0.12	0.12
MQL (ng/mL)	0.48	0.40	0.41
Spike recovery (%), *n* = 6			
0.5 (ng/mL)	84.4 ± 9.6	109 ± 8	110 ± 8
1 (ng/mL)	79.1 ± 9.8	100 ± 6	106 ± 10
5 (ng/mL)	81.2 ± 8.4	100 ± 4	119 ± 5
Matrix effect (%)	–14.4	13.2	22.2
Intra-day variation (%), *n* = 6			
0.5 (ng/mL)	8.83	10.8	10.1
1 (ng/mL)	3.13	3.19	3.46
5 (ng/mL)	7.18	9.10	4.61
Inter-day variation (%), *n* = 6			
0.5 (ng/mL)	9.09	7.51	12.9
1 (ng/mL)	9.25	5.93	10.6
5 (ng/mL)	7.22	7.85	6.61

^a^ The instrument linearity for all anlaytes in solvent (0.05–100 ng/mL). ^b^ The instrument linearity for all analytes in urine matrix (0.1–100 ng/mL).

The matrix effect is a common phenomenon in LC–MS analysis, especially in the ESI mode, that involves enhancement or suppression of analyte responses by matrix components [48]. We observed an ionization suppression for glyphosate (matrix effect: −14.4%) and ionization enhancements for AMPA (13.2%) and glufosinate (22.2%) (Table 2). The ionization suppression may explain the lower recoveries of glyphosate in fortified samples, which were in the range of 79.1–84.4%. However, the addition of labelled internal standards for quantification enabled correction for matrix effects.

We also validated our method by analyzing external quality assurance proficiency test (PT) urine samples, offered by the German External Quality Assessment Scheme (G-EQUAS) and the Quebec External Quality Assessment Scheme for Organic Substances in Urine (OSEQAS). Our results were within the acceptable ranges of assigned values, indicating high accuracy of our method (Table 3).

Next, we applied the validated method to the determination of concentrations in twenty human urine samples randomly collected from the populations of the US states of Iowa (*n* = 10) and New York (*n* = 10). In samples from Iowa, we detected glyphosate in six out of ten samples (mean: 1.18 ng/mL) and AMPA in five out of ten samples (mean: 0.88 ng/mL). In samples from New York, we found glyphosate in only one sample (0.53 ng/mL) and did not detect AMPA in any samples (Table 4). Samples from Iowa were collected from adult males living in a rural farming region, whereas those from New York were from a population of office workers including adult males and females. Glufosinate was not found in any of the samples, probably due to its rapid metabolism, and low usage (~200-fold lower than glyphosate) [1,2,17]. However, with the exponential increase in glufosinate usage, its concentration in human urine may increase in the future. Overall, our results suggest the feasibility of measuring glyphosate, AMPA, and glufosinate in biomonitoring studies using the current method.

In comparison to previous studies (Table 1), our method has several advantages for application in human biomonitoring studies: (1) Excellent chromatographic retention, resolution, and peak shape, which were achieved through the use of less corrosive (i.e., less acidic) mobile phases. Previous studies used very high concentrations of acids in mobile phases (i.e., 1% formic acid or acetic acid) [26,40], used ion-pairing reagents (i.e., heptafluorobutyric acid) [27,31], or applied derivatization steps [32] to enhance sensitivity and selectivity. Such techniques are tedious, time-consuming, or corrosive; (2) Our method provides excellent sensitivity and uses a smaller sample volume (250 µL) compared with other methods that used 0.5–2.5 mL urine [25,36]. Although one method reported relatively higher sensitivity [36], that method used a dilute-and-shoot method, which can affect selectivity and sensitivity due to matrix interferences. Furthermore, it was not clear if the reported detection limit for that method was that of the method or the IDL. (3) Our method has been validated through various QC parameters and successful participation in external assurance schemes while previously reported method did not report such external validation protocols (Table 3 and Table 4).

## 4. Conclusions

We have developed and validated an LC–MS/MS method for the determination of glyphosate, AMPA, and glufosinate in human urine. Passage of samples in sequence through a combination of cation- and anion-exchange solid-phase extraction cartridges for purification reduced matrix effects and increased sensitivity. Chromatographic and MS/MS methods were optimized for the separation of analytes from interferences as well as to improve peak shape and sensitivity. The analytical parameters, including linearity, sensitivity, accuracy, and precision, were suitable for trace analysis of glyphosate, AMPA, and glufosinate in human urine and for use in large-scale biomonitoring studies of exposure in the general population.

## Figures and Tables

**Figure 1 ijerph-19-04966-f001:**
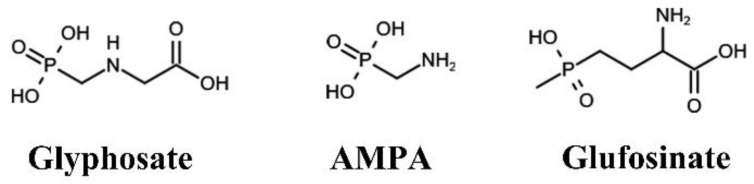
Molecular structures of the target analytes determined in this study. AMPA, aminomethylphosphonic acid.

**Figure 2 ijerph-19-04966-f002:**
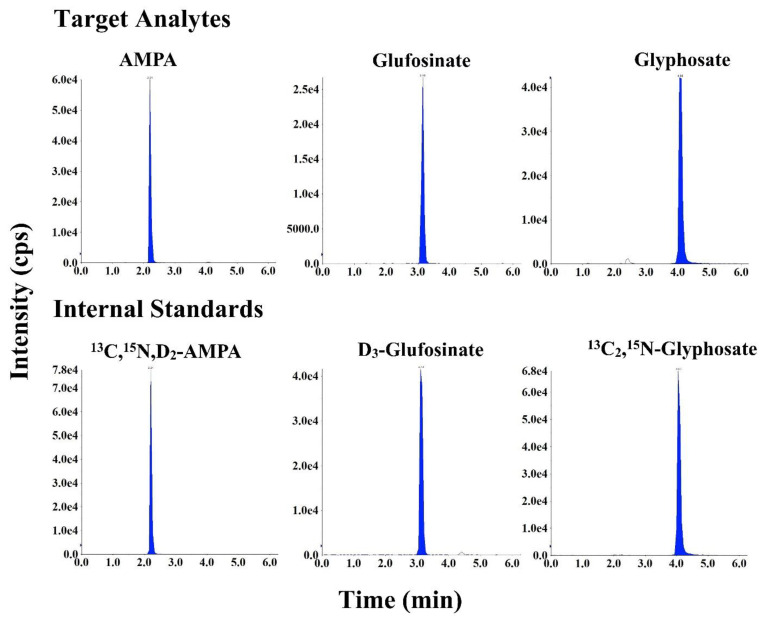
Representative MRM chromatograms of glyphosate, aminomethylphosphonic acid (AMPA), and glufosinate in neat standard solution (concentrations of the target analytes and internal standards were 10 ng/mL; injection volume was 20 µL).

**Figure 3 ijerph-19-04966-f003:**
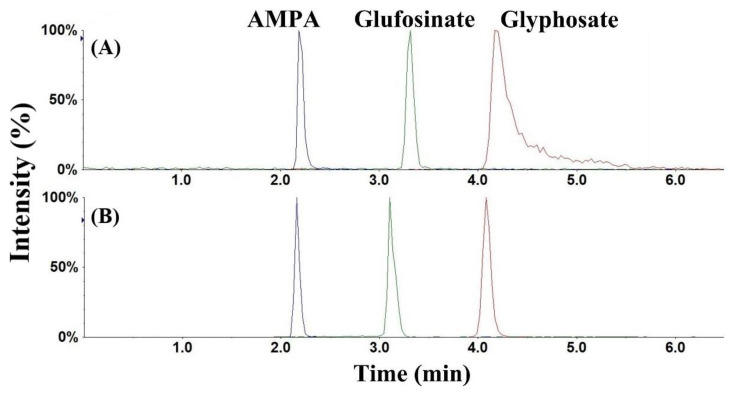
HPLC–MS/MS chromatograms of glyphosate, aminomethylphosphonic acid (AMPA), and glufosinate in neat standard solution before (**A**) and after (**B**) passivation of the LC system with medronic acid (analyte concentrations were 100 ng/mL; injection volume was 20 µL).

**Figure 4 ijerph-19-04966-f004:**
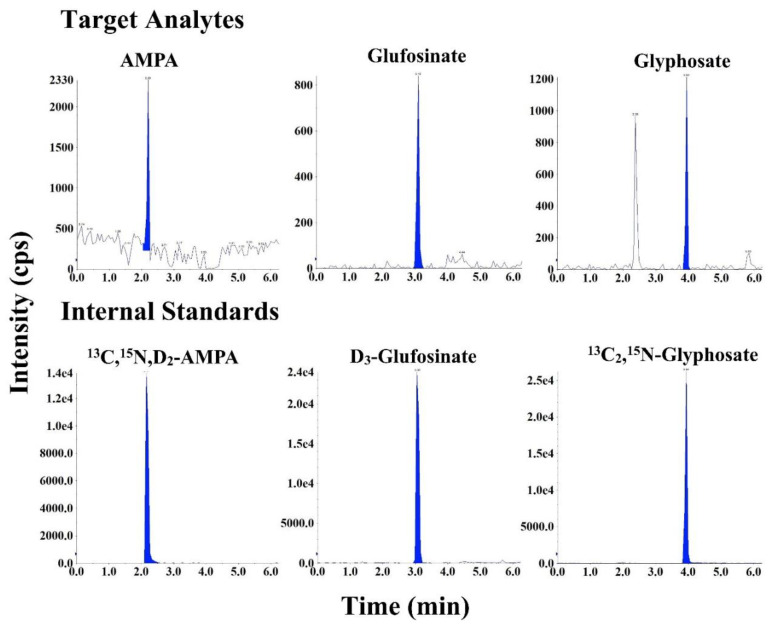
Representative MRM chromatograms of glyphosate, aminomethylphosphonic acid (AMPA), and glufosinate in pooled human urine spiked with 0.5 ng/mL native compounds and 10 ng/mL isotope-labelled internal standards (injection volume: 20 µL).

**Table 3 ijerph-19-04966-t003:** Glyphosate and AMPA concentrations measured in external quality assurance proficiency test urine samples using the method developed in this study and compared with the assigned values. Glyphosate and AMPA were assigned in the OSEQAS Round 2021-01 PT samples, while the G-EQUAS PT samples only include glyphosate. OSEQAS, Quebec External Quality Assessment Scheme for Organic Substances in Urine; G-EQUAS, German External Quality Assessment Scheme.

**OSEQAS Round 2021-01**						
ID	Glyphosate (ng/mL)			AMPA (ng/mL)		
	Assigned value	Acceptable range	Our results	Assigned value	Acceptable range	Our results
OS-U-E2101	1.24	0.713–1.77	1.55	1.65	0.954–2.35	1.65
OS-U-E2102	1.67	0.949–2.39	2.18	6.62	3.72–9.52	6.94
OS-U-E2103	2.23	1.24–3.22	2.80	2.2	1.31–3.09	2.15
**G-EQUAS Round 66/2020**						
ID	Glyphosate (ng/mL)					
	Assigned value	Acceptable range	Our results			
9A	0.64	0.49–0.79	0.78			
9B	1.2	0.93–1.47	1.37			

Abbreviations: AMPA, aminomethylphosphonic acid.

**Table 4 ijerph-19-04966-t004:** Glyphosate, AMPA, and glufosinate concentrations measured in twenty human urine samples randomly collected from the general populations in Iowa (*n* = 10) and New York (*n* = 10), USA. Calculated concentrations are provided for those between MDL and MQL.

ID	Location	Glyphosate (ng/mL)	AMPA (ng/mL)	Glufosinate (ng/mL)
1	Iowa, USA	0.54	0.50	<MDL
2	Iowa, USA	<MDL	<MDL	<MDL
3	Iowa, USA	0.91	0.39 (<MQL)	<MDL
4	Iowa, USA	3.04	1.21	<MDL
5	Iowa, USA	0.36 (<MQL)	0.44	<MDL
6	Iowa, USA	<MDL	<MDL	<MDL
7	Iowa, USA	0.70	0.85	<MDL
8	Iowa, USA	1.40	1.42	<MDL
9	Iowa, USA	0.49	0.19 (<MQL)	<MDL
10	Iowa, USA	0.27 (<MQL)	0.20 (<MQL)	<MDL
11	New York, USA	<MDL	<MDL	<MDL
12	New York, USA	<MDL	<MDL	<MDL
13	New York, USA	<MDL	<MDL	<MDL
14	New York, USA	<MDL	<MDL	<MDL
15	New York, USA	<MDL	<MDL	<MDL
16	New York, USA	<MDL	<MDL	<MDL
17	New York, USA	<MDL	<MDL	<MDL
18	New York, USA	<MDL	<MDL	<MDL
19	New York, USA	<MDL	<MDL	<MDL
20	New York, USA	0.53	0.39 (<MQL)	<MDL

Abbreviations: MDL, method detection limit; MQL, method quantification limit; AMPA, aminomethylphosphonic acid.

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
