# Peer review of "A Method for the Analysis of Glyphosate, Aminomethylphosphonic Acid, and Glufosinate in Human Urine Using Liquid Chromatography-Tandem Mass Spectrometry"

_ijerph, 2022, doi:10.3390/ijerph19094966_

Round 1

Reviewer 1 Report

Dear, Prof. Kannan,

This paper is interesting because of the development of pretreatment and analytical method of glyphosate, AMPA, and glufosinate in urine samples with external quality assurance PT urine samples using SPE with cation-exchange and anion-exchange cartridges and LC-MS/MS. This study itself seems scientifically sound, and acceptable for International Journal of Environmental Research and Public Health with polite and detailed discussions, but I think that this manuscript is necessary to be revised in minor scale mainly in rhetorical terms. I described several comments for the points I care :

Manuscript (ID: IJERPH-1673361)

General comments:

  This paper is interesting because of the development of pretreatment and analytical method of glyphosate, AMPA, and glufosinate in urine samples with external quality assurance PT urine samples using SPE with cation-exchange and anion-exchange cartridges and LC-MS/MS. This study itself seems scientifically sound, and acceptable for International Journal of Environmental Research and Public Health with polite and detailed analytical discussions, but I think that this manuscript is necessary to be revised in minor scale mainly in rhetorical terms.

Specific comments:

Abstract:

Page 1

Line 34: The abbreviation should be shown in Abstract because the abbreviation was used in the context as “proficiency test (PT)”.

Introduction:

Line 41: (hydroxyl (methyl)phosphonoyl)butanoic acid

    → (hydroxy(methyl)phosohoryl)butanoic acid(This notation is better.)

Page 2

Line 64: Please narrow the space after the period if the space is double-space.

Line 69: Please insert “for” between “evidence” and “genotoxicity”.

Results and discussion:

Page 5

Line 216: Table 1 appears before Table 2 in the context, so Table 1 itself should be shown before Table 2.

Line 231: 0.8­11.0 for glyphosate, 0.9­10.2 for AMPA, and 1.86­9.53 for glufosinate: It is better to describe only the values of low pKa for second dissociation constant related to anion forming among these ranges.

Page 6

Line 275: The description of formal name for “SPE”, “solid-phase extraction” is necessary as follows: solid-phase extraction (SPE)

Page 8

Table 2

Line 315-317: The abbreviations are not necessary because these are described in Line 318-320.

Line 320: Please modify the description as “limit. aThe” instead of “limit.aThe”.

Line 321: Please modify the description as “mL). bThe” instead of “mL).bThe”.

Page 9

Table 1

    Periods after “a” should be removed for the descriptions such as “MDL: 0.15a.” and “MDL: 0.1a.

    In “Human urine” on the bottom item, “D2,13C,15N-AMPA” is better instead of “13C,15N-D2-AMPA”.

Page 10

Line 336: Please modify the description as “chloride. aThe” instead of “chloride.aThe”.

Line 338: Please modify the description as “MDL/MQLs. bAnalytes” instead of “MDL/MQLs.bAnalytes”.

Line 339: The space is necessary between “mode.” and “Italicized”.

Line 339: The period is necessary for the last sentence as follows: quantitative ions monitored.

Line 344. Please confirm the value of ionization enhancement for AMPA (13.2%) calculated from spike recoveries (%) (109, 100, and 100 at 0.5, 1, and 5 ng/mL, respectively) on the basis of Equation 1: Matrix effect (%) = (A/B – 1) × 100

    where A and B are the slopes of analytes from the matrix-matched calibration curve and calibration curve prepared in neat solution, respectively.

Table 3

    Line 359: Please insert “;” between “Urine” and “G-EQUAS”.

Line 369: Adult males also for samples from office workers in New York?

Conclusions

Page 11

Line 403: Please correct the sentence as follows: “purification, reduced matrix effects, and ~” or “purification with reduced matrix effects and ~”

Author Response

Thank you very much for taking your time to review this manuscript. We really appreciate all your constructive comments which have enabled us to improve our work. We have revised our manuscript with all the changes highlighted. Appended in this letter is our point-to-point response to the comments raised by the reviewers. (Reviewer’s comments are in blue).

Reply to the 1st reviewer:

General comments: This paper is interesting because of the development of pretreatment and analytical method of glyphosate, AMPA, and glufosinate in urine samples with external quality assurance PT urine samples using SPE with cation-exchange and anion-exchange cartridges and LC-MS/MS. This study itself seems scientifically sound, and acceptable for International Journal of Environmental Research and Public Health with polite and detailed analytical discussions, but I think that this manuscript is necessary to be revised in minor scale mainly in rhetorical terms.

Reply: We appreciate the comments. The manuscript has been revised appropriately and the comments are replied below.

(1) Line 34: The abbreviation should be shown in Abstract because the abbreviation was used in the context as “proficiency test (PT)”.

Reply: Corrected (line 35)

(2) Line 41: (hydroxyl (methyl)phosphonoyl)butanoic acid → (hydroxy(methyl)phosophoryl)butanoic acid (This notation is better.)

Reply: Corrected (line 42).

(3) Line 64: Please narrow the space after the period if the space is double-space.

Reply: Corrected (line 65).

(4) Line 69: Please insert “for” between “evidence” and “genotoxicity”.

Reply: The sentence has been corrected (line 70).

(5) Line 216: Table 1 appears before Table 2 in the context, so Table 1 itself should be shown before Table 2.

Reply: Table 1 has been moved to before Table 2 in this revision.

(6) Line 231: 0.8­11.0 for glyphosate, 0.9­10.2 for AMPA, and 1.86­9.53 for glufosinate: It is better to describe only the values of low pKa for second dissociation constant related to anion forming among these ranges.

Reply: We agree with the reviewer. Only the lowest pKa values of glyphosate, AMPA, and glufosinate are provided in this revision (lines 240-242).

(7) Line 275: The description of formal name for “SPE”, “solid-phase extraction” is necessary as follows: solid-phase extraction (SPE)

Reply: Corrected (line 286).

(8) Line 315-317: The abbreviations are not necessary because these are described in Line 318-320.

Reply: Yes, the abbreviations are redundant, and have been deleted in this revision (lines 359-362 and lines 363-365).

(9) Line 320: Please modify the description as “limit. aThe” instead of “limit.aThe”.

Reply: Corrected (line 365).

(10) Line 321: Please modify the description as “mL). bThe” instead of “mL).bThe”.

Reply: Corrected (line 366).

(11) Table 1 Periods after “a” should be removed for the descriptions such as “MDL: 0.15a.” and “MDL: 0.1a.”

Reply: Corrected (Table 1).

(12) In “Human urine” on the bottom item, “D2,13C,15N-AMPA” is better instead of “13C,15N-D2-AMPA”.

Reply: Corrected (Table 1).

(13) Line 336: Please modify the description as “chloride. aThe” instead of “chloride.aThe”.

Reply: Corrected (line 346).

(14) Line 338: Please modify the description as “MDL/MQLs. bAnalytes” instead of “MDL/MQLs.bAnalytes”.

Reply: Corrected (line 348).

(15) Line 339: The space is necessary between “mode.” and “Italicized”.

Reply: Corrected (line 349).

(16) Line 339: The period is necessary for the last sentence as follows: quantitative ions monitored.

Reply: Corrected (line 349).

(17) Line 344. Please confirm the value of ionization enhancement for AMPA (13.2%) calculated from spike recoveries (%) (109, 100, and 100 at 0.5, 1, and 5 ng/mL, respectively) on the basis of Equation 1: Matrix effect (%) = (A/B – 1) × 100 where A and B are the slopes of analytes from the matrix-matched calibration curve and calibration curve prepared in neat solution, respectively.

Reply: The matrix effects were estimated using the Equation 1: Matrix effect (%) = *A/B – 1) *100%, where A and B are the slopes of analytes from the matrix-matched calibration curve and calibration curve prepared in neat solution, respectively (Table 1).

(18) Line 359: Please insert “;” between “Urine” and “G-EQUAS”.

Reply: Corrected (line 387).

(19) Line 369: Adult males also for samples from office workers in New York?

Reply: Yes. The sentence has been revised as “…a population of office workers including adult males and females.” (lines 397-398).

(20) Line 403: Please correct the sentence as follows: “purification, reduced matrix effects, and ~” or “purification with reduced matrix effects and ~”

Reply: The sentence has been corrected as “…for purification with reduced matrix effects and increased sensitivity” (line 431).

Reviewer 2 Report

This study described the development of an analytical LC-MS/MS method for the detection of glyphosate and derivatives in urine samples. It is a nice paper describing a comprehensive method development and is very well written. The analysis of glyphosate and derivatives is indeed challenging and roust methods of analysis, especially in complex matrices such as urine, are of great need and interest to evaluate human exposure and risk.

Some mostly minor comments and clarifications are below:

There are no mentions of ‘procedural blank’ QAQC samples.

Lines 26-17 – specify which analytes the “0.12–0.14 27 ng/mL” “0.40–0.48 ng/mL” MDLs refer to

Line 53 – The two references (4 and 5) to support glyphosate’s “ubiquitous presence” are North America based, consider adding additional global references.

Lines 57-58 – which of the analytes are “unchanged” in urine and faeces? Is one hypothesis not that glyphosate may potentially breakdown to AMPA? Do we really know enough about this transformation?

Do we know enough about excretion pathways and weather analysing urine only provides enough of a representation of exposure? Zoller et al. 2020 (Int J Hyg Environ Health, 228:113526 (doi: 10.1016/j.ijheh.2020.113526)) for example report that only 1% of glyphosate is excreted in urine. Of course this is not for this current paper do deal with this issue, as an analytical method paper, but something we may want to understand better when considering true exposure to glyphosate.

Lines 77-79 – Agree there is need for more study and little is available on human exposure, but interesting the authors highlight “especially in the general population” when from the evidence (studies) reported from various countries so far it seems the urinary concentrations in the general populations are quite low and arguably negligible. The ones of potentially more concern are from occupational cohorts with urinary concentrations up to ~ 30 times higher (Acquavella, J.F., et al.,Environmental health perspectives, 2004. 112(3): p. 321-326).

Line 131 – 250 ul of urine seems quite low. Why was this volume chosen?

Line 136 – so essentially the MCX is used as a “filter” step.

Line 135 – “The sample” what volume does this refer to? Is this the 250 ul of urine combined with the MQ:ACN = 0.1% Formic mixture? And what volume would this be in total?

Line 141 – describe the purpose of the NH4OH addition, and what is the volume of the ‘sample’ at this stage? See comment directly above.

Line 147 – interesting the evaporation to dryness, state the purpose/benefit here too.

Line 153 – Aren’t ‘Applied Biosystems’ called ‘Sciex’ or ‘AB Sciex’ these days?

Fig 2 – these are indeed nice chromatograms but what do they look like closer to the detection range that is required for low urine sample detection and in matrix? i.e. would be nice to see what a 0.5 / 1 ng/mL peak looks like in urine sample.

Lines 275-279 – Interesting that MCX worked better than HLB. Likely due to removal of sulfonic groups in the urine. Is there a way to assess quantitatively how much better / worse the other SPE sorbents performed compared to MCX?

Also wonder whether the extent of interferences could be urine / sample specific? So would you expect that the constituents / interferences would depend on the urine sample (i.e. differences in lifestyle / dietary composition etc. of individuals) and its concentration?

This is just a thought – no need to address it here.

Fig 4 – now that we see Fig 4, Fig 2 is probably redundant.

Author Response

General comments: This study described the development of an analytical LC-MS/MS method for the detection of glyphosate and derivatives in urine samples. It is a nice paper describing a comprehensive method development and is very well written. The analysis of glyphosate and derivatives is indeed challenging and roust methods of analysis, especially in complex matrices such as urine, are of great need and interest to evaluate human exposure and risk. Some mostly minor comments and clarifications are below:

Reply: We thank the reviewer for the comments. The specific comments are replied below.

(1) There are no mentions of ‘procedural blank’ QAQC samples.

Reply: Procedural blank (using water instead of urine) was used to monitor background contamination of the analytes. The information has been included in this revision (line 215-216).

(2) Lines 26-17 – specify which analytes the “0.12–0.14 27 ng/mL” “0.40–0.48 ng/mL” MDLs refer to.

Reply: The analytes have been specified in this revision (line 28).

(3) Line 53 – The two references (4 and 5) to support glyphosate’s “ubiquitous presence” are North America based, consider adding additional global references.

Reply: Additional reference has been added in this revision (line 54).

(4) Lines 57-58 – which of the analytes are “unchanged” in urine and faeces? Is one hypothesis not that glyphosate may potentially breakdown to AMPA? Do we really know enough about this transformation?

Reply: We appreciate this valuable comment. Glyphosate can be metabolized by microorganisms into AMPA in the environment, and AMPA has been detected in soils [1] and plants [2]. However, studies have reported that glyphosate and AMPA are poorly biotransformed in animals following administration [3]. AMPA can also originate from other sources, besides glyphosate.  In human body, only up to 0.3% of glyphosate are metabolized to AMPA [4, 5]. Glyphosate seems to accumulate principally in the kidneys, liver, colon, and small intestine and is eliminated in the feces (90%) and urine within 48 h [3]. Nevertheless, the urinary excretion fraction of glyphosate remains unclear. The European commission estimated that 30% of the absorbed glyphosate can be excreted through urine [6]. The European Food Safety Agency estimated a urinary excretion of 20% [3]. Recently, Zoller et al. reported a urinary excretion of 1% for glyphosate [5].  CDC has been using urine biomonitoring for glyphosate and therefore our focus was to measure these analytes in urine.

(5) Do we know enough about excretion pathways and weather analysing urine only provides enough of a representation of exposure? Zoller et al. 2020 (Int J Hyg Environ Health, 228:113526 (doi: 10.1016/j.ijheh.2020.113526)) for example report that only 1% of glyphosate is excreted in urine. Of course this is not for this current paper do deal with this issue, as an analytical method paper, but something we may want to understand better when considering true exposure to glyphosate.

Reply: Glyphosate is primarily eliminated in the feces and urine within 48 h [3]. The urinary excretion fraction of glyphosate remains unclear (1-30%) [3, 5, 6]. Urinary glyphosate seems represents a small fraction of the exposure. Previous human biomonitoring studies reported low detection frequencies and low concentrations of glyphosate and AMPA in urine samples, but the real exposure level might be much higher.  It is likely that other metabolites of glyphosate may be formed, but information is scant in that regard.

(6) Lines 77-79 – Agree there is need for more study and little is available on human exposure, but interesting the authors highlight “especially in the general population” when from the evidence (studies) reported from various countries so far it seems the urinary concentrations in the general populations are quite low and arguably negligible. The ones of potentially more concern are from occupational cohorts with urinary concentrations up to ~ 30 times higher (Acquavella, J.F., et al.,Environmental health perspectives, 2004. 112(3): p. 321-326).

Reply: Most of the humans are exposed to low levels of glyphosate and AMPA through food and water. However, long-term chronic exposure to low levels of such compounds may result in serious problems such as cancer [7]. Although urinary glyphosate concentration of general population is generally low (0.16–7.6 ng/mL) [8], the real exposure might be high given the low urinary excretion fraction of glyphosate (1–30%) [3, 5, 6].  Efforts to discover other metabolites of glyphosate warrant further studies.

(7) Line 131 – 250 ul of urine seems quite low. Why was this volume chosen?

Reply: We developed this method using a small sample volume (250 µL) to make it more applicable for human biomonitoring. This is because sample volume is always limited in large-scale human biomonitoring since there are various classes of analytes need to be analyzed. We were able to measure glyphosate, AMPA, and glufusinate using only 250 µL urine. Nevertheless, our method can be easily modified to use a larger volume (e.g., 500 µL) and even lower LODs/LOQs can be obtained.  Sometimes use of highere sample volume also means more purification could be required.  We thought that 250 uL is an optimal volume taken for extraction of several other environmental chemicals.  So, a compromise was to be made with regard to sample volume and extraction and purification steps to come to an optimal volume.

(8) Line 136 – so essentially the MCX is used as a “filter” step.

Reply: Yes, our results showed that using MCX as a pre-cleanup procedure can be efficient to improve sensitivity of the analytes, probably because of the efficient removal of cations by MCX cartridge.

(9) Line 135 – “The sample” what volume does this refer to? Is this the 250 ul of urine combined with the MQ:ACN = 0.1% Formic mixture? And what volume would this be in total?

Reply: “The sample” refers to urine sample fortified with internal standard. The internal standards and native standards were dissolved in water:ACN (95:5) containing 0.1% formic acid. The total volume was around 270 µL (250 µL urine + 10 µL internal standard + 10 µL native standard).

(10) Line 141 – describe the purpose of the NH4OH addition, and what is the volume of the ‘sample’ at this stage? See comment directly above.

Reply: NH4OH was added into the sample to improve the extraction recoveries. This is because under basic environment, the analytes are more readily extracted by MAX cartridges. The total volume was about 5 mL. The sentence has been revised (line 147).

(11) Line 147 – interesting the evaporation to dryness, state the purpose/benefit here too.

Reply: The elute (3 mL) was evaporated to dryness and redissolved in 250 µL of water:ACN (95:5) containing 0.1% formic acid. In this way, the analytes can be concentrated to aid the LC-MS/MS measurement.

(12) Line 153 – Aren’t ‘Applied Biosystems’ called ‘Sciex’ or ‘AB Sciex’ these days?

Reply: The name has been corrected in this revision (line 160).

(13) Fig 2 – these are indeed nice chromatograms but what do they look like closer to the detection range that is required for low urine sample detection and in matrix? i.e. would be nice to see what a 0.5 / 1 ng/mL peak looks like in urine sample.

Reply: Figure 4 shows the chromatograms in real urine sample spiked at 0.5 ng/mL.

(14) Lines 275-279 – Interesting that MCX worked better than HLB. Likely due to removal of sulfonic groups in the urine. Is there a way to assess quantitatively how much better / worse the other SPE sorbents performed compared to MCX?

Reply: We agree with the reviewer. Glyphosate can chelate with cations in the matrix and may reduce its recovery. MCX cartridges can efficiently remove cations, and therefore, we obtained improved sensitivity for glyphosate and glufosinate when using MCX as a pre-cleanup step (Fig. S1 and Table S2). The peak area and signal-to-noise ratio (S/N) could be used to quantitatively evaluate the performance of different cartridges.

(15) Also wonder whether the extent of interferences could be urine / sample specific? So would you expect that the constituents / interferences would depend on the urine sample (i.e. differences in lifestyle / dietary composition etc. of individuals) and its concentration? This is just a thought – no need to address it here.

Reply: We did not investigate the relationship between urine constituents and matrix effects in this study. However, an earlier study reported that glyphosate has a strong tendency to form glyphosate-metal complexes, especially with dications such as Ca2+, Mg2+, and Sr2+ in ocean water. The existence of such cations can significantly influence the chromatographic behavior and MS response of glyphosate [9]. We assume that cations in urine might be the source of such interferences.

(16) Fig 4 – now that we see Fig 4, Fig 2 is probably redundant.

Reply: We would like to present the chromatograms of the analytes in both neat solution and the urine matrix to show the efficiency of the optimized sample cleanup.

Reviewer 3 Report

The manuscript by Zhongmin Li  and Kurunthachalam Kannan reports a method for the determination of urinary herbicides (glyphosate, its metabolite AMPA and glufosinate)  largely used in agriculture and therefore spread in food and environment. Their potential toxicity has spurred the development of advanced analytical methods for large-scale human biomonitoring. The results presented in this study describes a liquid-chromatography-tandem mass spectrometry method that provides optimal chromatographic resolution of the sampled analytes, high sensitivity  and accuracy. A preliminary tretament of human urine with a combination of cation- and anion-exchange solid-phase -phase extraction cartridge has considerably increased the selectivity of the process and cut down matrix interferences.

The methodological approach is robust and the results are clearly described.

Overall, I think that the manuscript provides enough scientific insight to merit publication.

Minor comments:

Table 1 should be presented before Table 2 

Footnotes' numbers in all tables should be followed by round brackets.

Author Response

General comments: The manuscript by Zhongmin Li and Kurunthachalam Kannan reports a method for the determination of urinary herbicides (glyphosate, its metabolite AMPA and glufosinate) largely used in agriculture and therefore spread in food and environment. Their potential toxicity has spurred the development of advanced analytical methods for large-scale human biomonitoring. The results presented in this study describes a liquid-chromatography-tandem mass spectrometry method that provides optimal chromatographic resolution of the sampled analytes, high sensitivity and accuracy. A preliminary tretament of human urine with a combination of cation- and anion-exchange solid-phase -phase extraction cartridge has considerably increased the selectivity of the process and cut down matrix interferences. The methodological approach is robust and the results are clearly described. Overall, I think that the manuscript provides enough scientific insight to merit publication.

Reply: We thank the reviewer for the comments. The specific comments are replied below.

(1) Table 1 should be presented before Table 2

Reply: Corrected in this revision.

(2) Footnotes' numbers in all tables should be followed by round brackets.

Reply: In this revision, the numbers in footnotes have been changed as superscripts.

Reviewer 4 Report

The manuscript is dedicated to a problem of rapid and sensitive quantification of glyphosate and related compounds in urine which is of great practical importance for human biomonitoring. It is properly organized and well written. Authors proposed an LC-MS/MS technique combined with SPE sample cleanup and isotopically labelled standards to eliminate matrix interferences. The developed method is adequately validated and tested on real urine samples. I can recommend this high-quality work for publication in its presrnt form or after minor revision according to the comments below:

  1. Authors should estimate and report analyte recoveries in the developed sample cleanup procedure
  2. The use of wide-bore (4.6 mm) HPLC column and therefore high flowrate in ESI-MS with water-rich mobile phase looks unjustified due to the low efficiency of water spraying. Is there any reason to use such a column?

Author Response

General comments: The manuscript is dedicated to a problem of rapid and sensitive quantification of glyphosate and related compounds in urine which is of great practical importance for human biomonitoring. It is properly organized and well written. Authors proposed an LC-MS/MS technique combined with SPE sample cleanup and isotopically labelled standards to eliminate matrix interferences. The developed method is adequately validated and tested on real urine samples. I can recommend this high-quality work for publication in its present form or after minor revision according to the comments below:

Reply: We appreciate the reviewer’s comments.

(1) Authors should estimate and report analyte recoveries in the developed sample cleanup procedure.

Reply: The recoveries of the analytes in the developed procedure have been presented in Table 2. The recoveries spiked at 3 different levels (0.5, 1, and 5 ng/mL) ranged from 79.1% to 119%, with coefficients of variation (CVs) of 4–10%.

(2) The use of wide-bore (4.6 mm) HPLC column and therefore high flowrate in ESI-MS with water-rich mobile phase looks unjustified due to the low efficiency of water spraying. Is there any reason to use such a column?

Reply: We appreciate this valuable comment. There was not a specific reason. We compared the performance of different columns available in our lab, and found that the C6-Phenyl column provided the best chromatographic separation and peak shape for all analytes. However, we assume that a narrower column with less flow rate can be working and reduce the volume of mobile phases. We will test narrower columns in future studies.

Reviewer 5 Report

You investigated new protocol of measuring several substance in human urine. I didn’t understand final protocol which you investigated performance on line 382-398, so it is need to describe constitution of the measuring with figures.

How did you collect urine samples? Could you write how to get samples, and number of samples on 2.Materials and methods?

I felt your assessment of the measuring is enough, but description of your evaluation plan before 3.1 is needed. I think there is originality on 3.1, could you clearly write the point which you devised?

On line 267-269, it is nonobjective, is there reference?

On 4. Conclusions, you write HPLC-MS/MS, but you write LC-MS/MS on other section. You should be unification.

Author Response

(1) You investigated new protocol of measuring several substance in human urine. I didn’t understand final protocol which you investigated performance on line 382-398, so it is need to describe constitution of the measuring with figures.

Reply: This paragraph was to compare our method against previous ones, and show the advantages of our method in human biomonitoring. The final protocol of our method has been described with details in section 2.2 “Sample preparation” and section 2.3 “LC-MS/MS”.

(2) How did you collect urine samples? Could you write how to get samples, and number of samples on 2.Materials and methods?

Reply: Human urine samples used in this study were randomly selected from our previous study [10]. The information has been included in this revision (lines 128-131)

(3) I felt your assessment of the measuring is enough, but description of your evaluation plan before 3.1 is needed. I think there is originality on 3.1, could you clearly write the point which you devised?

Reply: The detection of glyphosate is challenging because of it is polar, highly hydrophilic, and has a small molecular weight. Previous studies tried different columns but the peak shape, retention, and chromatographic separation were always poor. In this study, we first compared columns with different mechanisms, and found that C6-Phenyl column can separate the three target analytes with good peak shape for AMPA and glufosinate. However, glyphosate had a tailing peak. We assumed this might be due to the chelation of glyphosate with metal ions in the HPLC hardware. Consequently, we passivated the LC system using medronic acid. After passivation, the C6-Phenyl column provided optimal performance for retention, separation, and peak shape for all analytes.

(4) On line 267-269, it is nonobjective, is there reference?

Reply: These were the results of our study. We found that using MAX cartridge only can provide good recoveries for all analytes at high spike levels (e.g., 10 ng/mL). However, the sensitivity was not good enough (Fig. S1) due to matrix effects. Using the combined procedure of MCX and MAX cartridges, we were able to obtain good recoveries and sensitivities (Table 2). This is essential because glyphosate concentration in human urine of general population is generally < 1 ng/mL.

(5) On 4. Conclusions, you write HPLC-MS/MS, but you write LC-MS/MS on other section. You should be unification.

Reply: Corrected (line 428).